# AHY-SLAM: Toward Faster and More Accurate Visual SLAM in Dynamic Scenes Using Homogenized Feature Extraction and Object Detection Method

**DOI:** 10.3390/s23094241

**Published:** 2023-04-24

**Authors:** Han Gong, Lei Gong, Tianbing Ma, Zhicheng Sun, Liang Li

**Affiliations:** 1State Key Laboratory of Mining Response and Disaster Prevention and Control in Deep Coal Mines, Huainan 232001, China; 2School of Mechanical Engineering, Anhui University of Science and Technology, Huainan 232001, China; 3School of Computer Science and Technology, University of Science and Technology of China, Hefei 230026, China; 4Suzhou Institute for Advanced Research, University of Science and Technology of China, Suzhou 215123, China

**Keywords:** visual SLAM, keyframe, dynamic scene, optical flow method, feature point

## Abstract

At present, SLAM is widely used in all kinds of dynamic scenes. It is difficult to distinguish dynamic targets in scenes using traditional visual SLAM. In the matching process, dynamic points are incorrectly added to the pose calculation with the camera, resulting in low precision and poor robustness in the pose estimation. This paper proposes a new dynamic scene visual SLAM algorithm based on adaptive threshold homogenized feature extraction and YOLOv5 object detection, named AHY-SLAM. This new method adds three new modules based on ORB-SLAM2: a keyframe selection module, a threshold calculation module, and an object detection module. The optical flow method is used to screen keyframes for each frame input in AHY-SLAM. An adaptive threshold is used to extract feature points for keyframes, and dynamic points are eliminated with YOLOv5. Compared with ORB-SLAM2, AHY-SLAM has significantly improved pose estimation accuracy over multiple dynamic scene sequences in the TUM open dataset, and the absolute pose estimation accuracy can be increased by up to 97%. Compared with other dynamic scene SLAM algorithms, the speed of AHY-SLAM is also significantly improved under a guarantee of acceptable accuracy.

## 1. Introduction

As early as in 1986, Randall C. and Peter Cheeseman proposed the Simultaneous Localization And Mapping (SLAM) problem [1], while SLAM is still an unsolved technical problem. Then, Leonard and Durrant Whyte solved the SLAM problem by estimating the robot pose [2]. Smith et al. [3] used Kalman filtering to improve SLAM and make it more widely used. However, due to the limitations of computer technology, SLAM has not received public attention. After the year 2000, with the rapid development of computer technology, there was no technical limit, and image processing technology quickly became the focus of research. Then, visual SLAM began to enter the public vision, and after decades of development, there are some very mature VSLAM algorithms, such as PTAM [4], LSD-SLAM [5], DSO [6], ORB-SLAM2 [7], VINS Mono [8], and RGBD-SLAM [9]. However, these SLAM algorithms assume that objects in the environment are static or moving at low speeds, and the processing images do not distinguish dynamic objects in scenes. The presence of dynamic objects, such as people and cars, in real-world scenarios is inevitable. A lot of SLAM algorithms use feature points, and since feature points are selected more often in areas with strong textural information [4,7,8,9], if a dynamic object has strong textural information, the systems will extract a large number of features from the dynamic object. When the unsteady feature points are tracked, the attitude estimation will be seriously affected, resulting in a large trajectory error, or even tracking loss, and the map constructed with the dynamic object will also appear with a large number of double shadows. Some optimization algorithms, such as random sample consensus (RANSAC) and graph optimization [10], can filter out a small number of weak dynamic features in dynamic scenes as outliers, but they cannot filter out the influence of dynamic targets in the face of a large proportion of dynamic targets or fast-moving dynamic scenes, causing positioning misalignment.

ORB-SLAM2 is a classic, improved open-source visual SLAM solution based on ORB-SLAM [11] developed by the University of Zaragoza in Spain in 2015. It is the most representative method in feature point SLAM. In recent years, based on ORB-SLAM, a lot of studies have emerged around dynamic scenes [12,13,14]. Among them, their aim is mainly to eliminate dynamic feature points by constructing relevant methods with semantic constraints through deep learning. Semantic constraints are mainly divided into semantic segmentation and target detection to obtain semantic information in the environment. Semantic segmentation can provide better pixel-level object masks, but it is less in real-time. Although a pixel-level segmentation area could be obtained, it would consume a lot of computing power, the final segmentation boundary would not be a complete object boundary, and there would still be many misjudgment points [15]. Object detection can quickly obtain detection boxes for different objects. Since it does not need to judge each pixel point, it is much better than semantic segmentation in terms of real-time performance. SLAM has high requirements for real-time performance and cannot provide enough computing power to support semantic segmentation on most mobile platforms, so object detection algorithms are widely used in lightweight scenario SLAM [16].

However, the ORB-SLAM2 algorithm combined with object detection still has the following problems in terms of processing efficiency and detection accuracy:

(1) Object detection uses detection boxes when labeling different objects. Due to the uneven distribution and serious aggregation of ORB feature points in an image, it often occurs that the feature points in some frames are concentrated in the detection box of a dynamic object, which leads to a small number of remaining feature points after deleting the dynamic points, affecting the matching accuracy.

(2) The detection box is not the actual boundary of an object. If all dynamic feature points in a block diagram are deleted directly, some static feature points will be mistakenly deleted.

(3) The detection framework used with ORB-SLAM2 needs to extract ORB feature points for each frame and then screen out the keyframes. Considering that the proportion of keyframes is low, and the process of calculating FAST corners and BRIEF descriptors frame by frame is time-consuming, a large number of redundant calculation processes lead to the algorithm having a low overall operational efficiency. The above problems seriously restrict the applicability of the object detection ORB-SLAM2 algorithm in a real environment.

To solve the above problem 1, this paper proposes an adaptive threshold homogenized feature extraction algorithm. The algorithm calculates the corner extraction threshold separately according to the gray-scale value of each small area in an image instead of using the global threshold, which can effectively disperse heavily aggregated ORB feature points evenly to most areas of the image. The problem of information redundancy caused by feature point aggregation is solved.

For problem 2, this paper classifies all detection objects according to dynamic degree and adopts different processing methods for feature points in detection boxes with different dynamic degrees so that the feature points are divided into dynamic feature points and static feature points. By eliminating the dynamic feature points in detection boxes and retaining the static feature points, the possibility of feature points being deleted by mistake is reduced.

Given problem 3, this paper improves the original keyframe selection method of ORB-SLAM2, using a more rapid optical flow method to process all incoming ordinary frames in advance to judge whether a frame is the keyframe needed by the system and only carrying out ORB feature extraction for keyframes, avoiding the consumption of descriptor calculation for each frame.

The main contributions of this paper are as follows:

(1) A dynamic scene visual SLAM algorithm based on homogenized feature extraction and the object detection method is proposed in this paper: AHY-SLAM (adaptive threshold and homogenized feature extraction and YOLOv5). This new method adds three new modules to ORB-SLAM2: a keyframe selection module, a threshold calculation module, and an object detection module. Moreover, homogenized ORB feature point extraction and dynamic feature point elimination steps are added to the tracking thread, improving the estimation accuracy and robustness of ORB-SLAM2 in dynamic scenarios.

(2) A keyframe selection method based on the optical flow method is proposed. The original operating framework of ORB-SLAM2 is modified, the LK optical flow method is introduced to track and locate ORB feature points before calculating them, and a new keyframe selection strategy is proposed to filter the keyframes. Only keyframes can extract ORB feature points, reducing the computing power consumption of the ORB feature points for each frame.

(3) An adaptive threshold homogenized feature extraction algorithm is proposed, which can adjust the detection threshold of FAST corners through the gray-scale values of different regions. Compared with the ORB feature point method using a fixed threshold, the distribution uniformity of feature points is improved and the possibility of feature points being too concentrated on dynamic targets is avoided.

(4) AHY-SLAM was verified in the TUM dataset. Compared with ORBSLAM2, the accuracy and robustness of AHY-SLAM in dynamic scene sequences are greatly improved, and the absolute trajectory error can be reduced by up to 97%. Compared with other dynamic scene SLAM algorithms [17,18,19], AHY-SLAM also achieves the best localization accuracy in multiple sequences and is faster.

## 2. Related Work

### 2.1. SLAM Methods in Dynamic Scenes

At present, the SLAM methods used in dynamic scenes mainly have two branches: the methods based on the traditional recognition algorithm and the methods based on deep learning.

Traditional dynamic SLAM relies on geometric constraints to separate static and dynamic features, and dynamic elements violate the standard constraints defined for static scenes in multi-view geometry. Kundu et al. [20] detected dynamic features through polar geometry and FVB constraints. Zou et al. [21] used reprojection errors to determine dynamic features by measuring the distances between tracked feature projections. Tan et al. [22] projected keyframes onto the current frame to detect features that had changed in appearance and structure and used the adaptive RANSC algorithm to remove abnormal features. Wang et al. [23] carried out feature matching between two adjacent frames through polar constraints and then conducted clustering on depth images to obtain dynamic objects. In addition to the method of using geometric constraints, the optical flow method is also often used for dynamic feature recognition. Klappstein et al. [24] defined the possibility of moving objects in a scene according to the motion measure calculated from optical flow. Derome et al. [25,26] used the residual difference between a predicted image and a binocular camera observation image to calculate optical flow, and the predicted image was calculated by estimating the camera motion to convert the current frame to the previous frame. Moving objects were then observed by detecting points in the residual field. Wang et al. [27] completed the segmentation of moving objects through optical flow calculation, sparse point trajectory clustering, and densification operations, in turn, and used model selection to cope with the situation of over-segmentation or under-segmentation.

With the rapid development of deep learning in recent years, object detection and semantic segmentation have been greatly improved, both in accuracy and speed. Many people have begun to consider the introduction of deep learning into SLAM. In the running process, outliers in moving objects are eliminated with detection boxes or pixel-level segmentation to exclude the influence of dynamic features on tracking and positioning and make the final map without the shadow interference of moving objects so as to build a more accurate map of the environment. Bescos et al. [17] proposed DynaSLAM, and this SLAM system used MaskR-CNN [28] for the semantic segmentation of input images for the first time and then used multi-view geometry to eliminate dynamic objects in the images. Zhong et al. [18] proposed Detect-SLAM, which is similar to DynaSLAM, wherein the SLAM system uses SSD object detection neural networks to identify objects in images and then propagates the motion probability with feature matching, increasing the area of influence to remove the influence of dynamic points in the environment. Yu et al. [19] proposed a DS-SLAM algorithm to remove the influence of dynamic features by combining semantic segmentation networks with optical flow methods. Wu et al. [29] used the YOLOv3 object detection algorithm [30] to identify dynamic targets and eliminated the influence of dynamic targets by eliminating outliers in detection boxes.

### 2.2. Improved Extraction of Homogenized ORB Feature Points

The visual odometer in SLAM based on the feature point method needs to select the same representative points from each image. ORB feature points are widely used in visual SLAM systems because of their good real-time performance, which is 10 times faster than that of SURF feature points and 100 times faster than that of SIFT feature points [31]. However, it also has some obvious defects, such as uneven distribution and serious aggregation of extracted feature points in an image, resulting in poor accuracy in the subsequent image matching and large deviation in the camera pose estimation results. Many researchers have improved the ORB detection algorithm. Mur-Artal et al. [7] used the quadtree splitting algorithm to divide an image into regions, evenly distribute feature points to these regions, and perform non-maximum suppression of feature points by calculating the Harris response value [32]. Yu et al. [33] improved the traditional quadtree algorithm based on Raul, adding regional edge detection and feature point redundancy to improve the uniformity of feature points. Sun et al. [34] set an adaptive threshold according to extracted feature points and shortened the time of feature extraction by judging whether the total number was up to standard.

## 3. Materials and Methods

### 3.1. System Architecture

The system architecture of AHY-SLAM is shown in Figure 1. The blue box is the innovative part of AHY-SLAM compared with ORB-SLAM2, which includes three new modules, namely, keyframe selection, threshold calculation, and object detection. Homogenized ORB feature point extraction and dynamic feature point elimination steps are added to the tracking thread.

Each ordinary frame entering the system is first processed in two modules: keyframe selection and object detection. After an ordinary frame enters the keyframe selection module, AHY-SLAM extracts the FAST corners of the image and tracks them using the LK optical flow method. According to the improved keyframe decision, whether the ordinary frame can be selected as a keyframe is judged. For non-keyframes that do not meet the requirements, AHY-SLAM uses PNP + RANSAC to calculate the camera pose and continue to judge the next frame. If the ordinary frame is judged to be a keyframe, it is output to the threshold calculation module, wherein the image is divided into several small regions, and the corner calculation threshold is calculated separately according to the grey value of a region. All thresholds are exported to the homogenized ORB feature point extraction step in a tracking thread.

On the other side, after the ordinary frame is input into the object detection module, AHY-SLAM uses YOLOv5 to detect an object and draw a detection box around the detected object. For objects with different levels of dynamics, AHY-SLAM classifies them. Finally, the detection box positions of the different objects are output to the tracking thread for dynamic feature point elimination.

### 3.2. Keyframe Selection Module Based on Optical Flow Method

The information redundancy between similar frames acquired with the camera is relatively high, which has a limited effect on the accuracy improvement of the system. Moreover, if all frames are put into the local map construction and closed-loop detection thread, a large number of redundant calculations will be caused. To solve this problem, ORB-SLAM2 adopts a way of selecting keyframes that involves dividing all ordinary frames into two categories: keyframes and non-keyframes. The camera receives one frame each time (a stereo or RGB-D receives two images) to form one frame. The keyframe is filtered from the ordinary frame, which can only be used to solve the camera pose and tracking state of the current frame, while the keyframe can be used to determine the loopback detection and complete the local map construction. In addition, normal frames only resolve the current frame situation, while keyframes can have an impact on the entire system.

Because the keyframe selection strategy adopted in the ORB-SLAM2 system needs to obtain the feature point data on an image in advance, it is necessary to extract an ORB feature point from every incoming frame image. The ORB feature point method needs to extract FAST corners from images and calculate BRIEF descriptors and then find feature matching in different images by calculating the distances of descriptors. Therefore, extracting ORB feature points from all incoming ordinary frames would be computationally intensive and time-consuming for the entire system. ORB-SLAM2 optimizes a keyframe, but a keyframe is only a small fraction of the total frames, so it cannot achieve a significant optimization effect in actual use.

To improve this situation, AHY-SLAM implements a keyframe selection module based on the optical flow method. This module mainly uses the LK optical flow method to judge keyframes in advance so as to avoid complicated descriptor calculation.

Optical flow, a concept first proposed by Stoffregen et al. [35], carries information, such as about the direction and amplitude of moving objects. It is a method for describing the motion of pixels between images over time, essentially estimating the motion of pixels in images at different times. Different from the feature point method, the optical flow method does not use descriptors to find matching points, but directly uses gray information between images to track corners, so the calculation amount is much less than that for the feature point method. Optical flow is usually dense and sparse, and the dense optical flow method is represented in the HS optical flow method, and the sparse optical flow law is represented in the LK optical flow method [36]. Since only the optical flow method is needed to track feature points and corners in SLAM, the sparse optical flow method is generally used. AHY-SLAM avoids the overhead of constructing descriptors and matching by introducing LK optical flow into the keyframe selection module to directly track corners. The keyframe selection module extracts FAST corners from the first normal frame that is passed in and uses the LK optical flow method to track the corners in the first frame in subsequent frames. Its tracking mode is shown in Figure 2. The red boxes in the figure represent the position of a FAST corner in different frames.

The optical flow method tracks corners according to the assumption of gray-scale value invariance and motion constraints. Letting the gray-scale value of a pixel at (x,y) at time *t* be I(x,y,t), and moving it to (x+dx,y+dy) at time t+Δt, its gray-scale value is
(1)I(x+dx,y+dy,t+dt)

The following can be obtained from the assumption of gray-scale value invariance:(2)I(x+dx,y+dy,t+dt)=I(x,y,t)

The gray-scale value at time t+Δt is expanded via Taylor, and the first-order term is retained to obtain the following:(3)$$I(x+dx,y+dy,t+dt)\approx I(x,y,t)+\frac{\partial I}{\partial x}dx+\frac{\partial I}{\partial y}dy+\frac{\partial I}{\partial t}dt$$

Via gray-scale value invariant reduction, both sides are obtained:(4)$$\frac{\partial I}{\partial x}dx+\frac{\partial I}{\partial y}dy+\frac{\partial I}{\partial t}dt=0$$
(5)$$\frac{\partial I}{\partial x}dx+\frac{\partial I}{\partial y}dy=-\frac{\partial I}{\partial t}dt$$
where $\frac{\partial I}{\partial x}$ represents the gradient $I_{x}$ of the image in the direction of x; $\frac{\partial I}{\partial y}$ represents the gradient $I_{y}$ of the image in the direction of y; $\frac{\partial x}{\partial t}$ represents the gradient u of the image in the direction of x; $\frac{\partial y}{\partial t}$ represents the gradient v of the image in the direction of y; and $\frac{\partial I}{\partial t}$ represents the change $I_{t}$ of image gray-scale value with respect to time t.

The final result is
(6)$$I_{x}u+I_{y}v=-I_{t}$$

The above equation is a binary first-order equation, which cannot be solved by using one pixel point and needs to refer to additional constraints. That is, it is not only assumed that the gray-scale value of pixels is constant but also that the motion mode of pixels in the same window is the same.

Setting a window of size w×w, it then contains $w^{2}$ pixels altogether, and multiple equations can be obtained:(7)$${\left[\begin{array}{ll} I_{x} & I_{y} \end{array}\right]}_{k}\left[\begin{array}{c} u\\ v \end{array}\right]=-I_{tk},\;k=1,\ldots ,w^{2}$$

By solving the above equation, we obtain
(8)$$\left[\begin{array}{l} u\\ v \end{array}\right]={\left[\begin{array}{cc} \sum I_{x_{i}}^{2} & \sum I_{x_{i}}I_{y_{i}}\\ \sum I_{x_{i}}I_{y_{i}} & \sum I^{2}y_{i} \end{array}\right]}^{-1}\left[\begin{array}{c} -\sum I_{x_{i}}I_{t_{i}}\\ -\sum I_{y_{i}}I_{t_{i}} \end{array}\right]$$

Since the optical flow method is a nonlinear optimization problem, it is assumed that the initial value of optimization is close to the optimal value to ensure the convergence of the algorithm to a certain extent. If the camera motion is fast and the difference between two images is obvious, then the single-layer image optical flow method can easily reach a local minimum. AHY-SLAM addresses this problem by introducing an image pyramid. The lowest layer of the pyramid, that is, of the number of layers in the original image, is the 0th layer. The original image is constantly reduced to reduce the image resolution. When the resolution of the top layer image is reduced to a certain extent, the movement of pixels becomes small enough to meet the prerequisite of a small movement in the LK optical flow method. The optical flow is estimated from the top layer of the pyramid, and then it is iteratively calculated layer by layer along the pyramid structure to constantly correct the displacement assumed at the beginning so as to obtain the optical flow motion estimation for the original image. The advantage of moving from coarse to fine layers of optical flow is that while the pixel motion of the original image is large, the motion remains within a small range from the top of the pyramid.

After using optical flow to track the corners, AHY-SLAM sets the following new keyframe selection criteria based on the original steps of ORB-SLAM2:Whether the number of frames in the distance from the last keyframe is enough, that is, the keyframe cannot take into consideration a frame with a similar time;Whether the movement distance of the last keyframe is far enough, that is, the keyframe cannot take into consideration a frame with less movement, and rotation and displacement can be considered at the same time;Whether there is any difference in the number and proportion of co-view points compared with the last keyframe, that is, when the camera is facing the same scene, no repeated keyframes are recorded, and a new keyframe is created only when the camera leaves the scene.

In addition to the above three selection criteria, because the optical flow method itself has weaker adaptability to environmental changes than the feature point method, it needs to be considered that the optical flow method cannot track enough corners for camera pose calculation. As shown in Figure 3a, the number of corners tracked using the optical flow method may be insufficient. To this end, AHY-SLAM adds a keyframe selection condition:4.When the total number of corners tracked using the optical flow method does not meet the requirements, a frame is selected as a keyframe, and the image after ORB feature extraction is re-entered into the keyframe selection module as the new first frame for follow-up tracking. Figure 3b shows the result of extracting ORB feature points from the keyframe. It can be seen that more points are extracted, and these points will be tracked using the optical flow method.

The overall process of the keyframe selection module is shown in Algorithm 1. After the first frame is passed into the module, FAST corners are extracted, and then these corners are tracked using the LK optical flow method. Whether this frame can be selected as a keyframe is judged according to the above four keyframe judgment criteria. For non-keyframes, AHY-SLAM uses PNP + RANSC to predict the camera pose and then continues to repeat the above judgment process for the next frame. If the frame is selected as a keyframe, AHY-SLAM will extract ORB feature points from it, and the keyframe will become the new first frame and be tracked using LK optical flow method to repeat the judging process. By replacing the ORB feature point method with the optical flow method for keyframe judgment, unnecessary computing costs for non-keyframes can be saved.
**Algorithm** **1:** Keyframe selection process
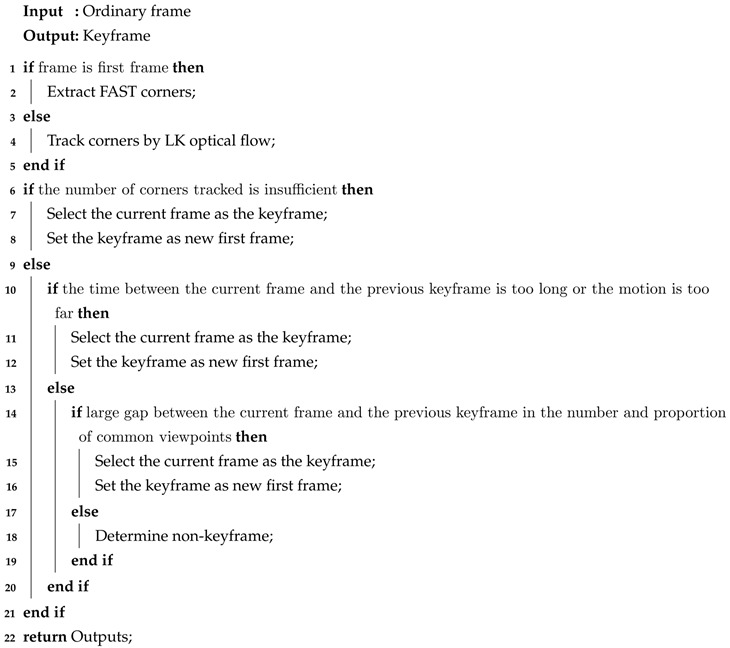


### 3.3. Adaptive Threshold Calculation Module and Homogenized ORB Feature Point Extraction

ORB is one of the fastest and most stable feature point detection and extraction algorithms. ORB feature points are composed of FAST corners and BRIEF descriptors. A FAST corner is a fast detection corner, which is faster than other corner detection algorithms because it only needs to compare the gray-scale differences between the center point and 16 pixel points within a radius of 3.

As shown in Figure 4, the main detection process for the central pixel *P* is as follows:Record the gray-scale value of the central pixel point as *I_p_*;Set an appropriate threshold *t*;With this pixel as the center point, select the discretized Bresenham circle with a radius of 3 pixels, and select 16 pixel points on the boundary of the circle;Compare the gray-scale values of the 16 pixel points with *I_p_*. When the gray-scale values of *n* consecutive pixel points (usually 12 or 9) are greater than *I_p_* + *t* or less than *I_p_* − *t*, the point is a corner.

Because it takes a lot of time to go through the detection of 16 pixel points, the method of rapid detection is usually adopted for only a few points to improve efficiency. For example, FAST corner detection in ORB-SLAM2 is used to detect only the gray-scale differences between pixels 1, 9, 5, and 13 of the 16 pixel points and the center point. Whether 3 or more of them exceed the threshold is checked. If so, they are selected as alternative corner points for subsequent detection.

The problem with selecting FAST corners in this way is that the set threshold *t* is an arbitrary constant value. In this way, only the point with the most obvious gray-scale difference in the whole image is extracted as a corner. In the subsequent quadtree partition extraction, there will be a large number of partition areas without corners. The method of reserving the maximum Harris response value according to quadtree screening eventually leads to all corners being concentrated in the area with the biggest change in light and shade in the whole image, forming aggregation. For the whole SLAM system, the excessive aggregation of feature points produces a large number of redundant feature point information, which will have a certain impact on the matching progress of subsequent images and the positioning accuracy of the camera. For SLAM that uses dynamic points elimination, this is more serious. If dynamic objects account for a large proportion of an image, it is easy for feature points to gather in the dynamic region, resulting in a small number of static points remaining after dynamic points elimination. If you end up keeping too few static points, you can lose trace at worst, which can seriously affect the functioning of the entire SLAM system. The larger the set threshold *t* is, the more serious the situation becomes.

To solve the above problems, AHY-SLAM sets up the adaptive threshold calculation module to independently calculate the threshold for each small region of keyframes and adjust it according to the number of extracted feature points. The thresholds obtained with the adaptive threshold calculation module are used in the homogenized ORB feature point extraction step. The overall process of the adaptive threshold calculation and homogenized ORB feature point extraction is shown in Algorithm 2. This module receives keyframes from the output of the keyframe selection module. To ensure that the extracted feature points have scale invariance, the image pyramid needs to be built on the keyframe first, and the number of layers of the pyramid needs to be determined according to the size of the image. The strategy for extracting feature points at different levels involves spreading all feature points evenly on each layer of the image in proportion to the image area. Assume that the total number of extracted ORB feature points is *N*, the total number of scaled pyramid layers is *m*, the height of the original image is *H*, the width is *W*, the image area is *W* × *H* = *C*, and the image pyramid scaling factor is *s*
(0<s<1).

The total image area of the image pyramid can be expressed as
(9)$$\begin{array}{cc} S & =H\cdot W\cdot {\left(s^{2}\right)}^{0}+H\cdot W\cdot {\left(s^{2}\right)}^{1}+\cdot \cdot \cdot +H\cdot W\cdot {\left(s^{2}\right)}^{m-1}\\ & =HW\frac{1-{\left(s^{2}\right)}^{m}}{1-s^{2}}=C\frac{1-{\left(s^{2}\right)}^{m}}{1-s^{2}} \end{array}$$

The number of feature points assigned per unit area is
(10)$$N_{avg}=\frac{N}{S}=\frac{N}{C\frac{1-{\left(s^{2}\right)}^{m}}{1-s^{2}}}=\frac{N\left(1-s^{2}\right)}{C\left(1-{\left(s^{2}\right)}^{m}\right)}$$

The number of feature points assigned via the original image layer is expressed as
(11)$$N_{0}=\frac{N\left(1-s^{2}\right)}{1-{\left(s^{2}\right)}^{m}}$$

The number of feature points assigned in layer *i* is expressed as
(12)$$N_{i}=\frac{N\left(1-s^{2}\right)}{C\left(1-{\left(s^{2}\right)}^{m}\right)}C{\left(s^{2}\right)}^{i}=\frac{N\left(1-s^{2}\right)}{1-{\left(s^{2}\right)}^{m}}{\left(s^{2}\right)}^{i}$$

After the image pyramid is constructed, meshes are divided for each layer according to their size. Independent detection thresholds are set for each divided mesh, and their initial values are set according to the gray-scale value in each mesh. A divided threshold *t* is expressed as
(13)$$t=\frac{\lambda }{m}\sum_{j=1}^{m}\frac{{\left[I\left(x_{j}\right)-\bar{I(x)}\right]}^{2}}{\bar{I(x)}}$$

In the above equation, *λ* is the scale factor, which is generally 0.01; *m* is the number of pixels in the mesh; and $I\left(x_{j}\right)$ and I(x) are the gray-scale value for each pixel and the average gray-scale value of the mesh, respectively.

After mesh division, feature extraction is carried out in the first layer of the image pyramid by traversing the mesh according to the threshold value *t* obtained with each mesh. In the extraction process, the feature points extracted from all the meshes of each layer are counted. If the total number is greater than or equal to *N_i_*, the loop is withdrawn to end the extraction of feature points from this layer. In some cases, the number of feature points extracted from a layer using this threshold value does not meet the requirements of *N_i_*. In this case, the threshold value is adjusted, and all the original extracted points are retained, while the threshold value is reduced to *t*/2, and the extraction is stopped until the total number of points in this layer meets the requirements.

After all feature points are extracted, the total number of feature points may be large, and they are still densely distributed in some areas. Therefore, quadtree splitting should be further carried out to delete redundant points in the image after the feature points are extracted. The specific method is as follows: First, set the total number of feature points *M* to be retained, divide the input image into 4 nodes, and gradually judge the number of feature points contained in each node. If the number of feature points in a node is greater than 1, further split it into 4 nodes; if it is equal to 0 or 1, do not split it anymore. Calculate the total number of nodes *m* to be split. If the total number of nodes *m* is greater than or equal to *M*, stop all splitting operations. If there are still multiple feature points in a node, perform non-maximum suppression according to the Harris response value.
**Algorithm** **2:** Adaptive threshold calculation and homogenized ORB feature point extraction process
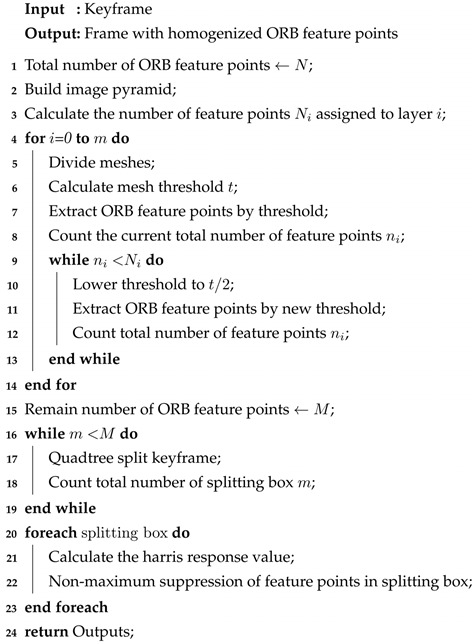


Considering that it is impossible to obtain all images with high quality from the environments faced in SLAM [37,38,39], it is necessary to conduct experiments on images of different qualities in the image dataset to confirm whether the uniformity of the new method in feature extraction meets requirements [40,41,42]. The ORB feature points extracted using the proposed method and the traditional ORB algorithm, separately, were compared in the Mikolajczyk open-source image dataset [43], and the effects are shown in Figure 5.

In order to test the effect of this algorithm on the uniformity of feature points, the uniformity calculation method proposed by Yao et al. [44] in the literature is adopted. An image is divided into regions in the vertical, horizontal, and 45° and 135° directions as well as in the center and periphery. The number of feature points in each region is counted separately and is denoted as the regional statistical distribution vector. The variance *V* of the vector and the uniformity *u* are calculated as follows:(14)u=−101lgV

It can be seen in Table 1 that the improved method greatly improves the uniformity of feature point extraction, and the average uniformity is increased by 23.8%, which solves the problem of redundancy caused by the excessive aggregation of ORB feature points. Based on the TUM dynamic scene dataset, we further verified the effect of uniformity enhancement on removing dynamic feature points and retaining static feature points in dynamic scenes. Figure 6 shows a frame in the freiburg3_walking_halfsphere sequence. The red points in the image are dynamic feature points, while the green points are static feature points. Figure 6a is the processing result without using homogenized extraction; Figure 6b is the processing result after using the homogenized extraction method in this paper. It is obvious that a large number of feature points in Figure 6a are concentrated on the dynamic objects, while the remaining static points are very few. If all dynamic points are eliminated, the accuracy of the system will be greatly affected. In contrast, Figure 6b retains a large number of feature points outside the dynamic objects, so it is necessary to carry out homogenized extraction of the feature points in the dynamic scene.

### 3.4. Object Detection Module Based on YOLOv5 and Dynamic Feature Point Elimination

As ORB-SLAM2 itself has a weak ability to deal with dynamic objects, it can only remove partial dynamic points on small objects through RANSC and cannot deal with scenes with a large number of dynamic objects. In order to make the system suitable for dynamic scenes, AHY-SLAM sets an object detection module base on YOLOv5. The detection results were applied to the step of dynamic feature point elimination. Algorithm 3 shows the overall process of the object detection module base on YOLOv5 and dynamic feature point elimination. After the RGB images of each frame are input into the system, AHY-SLAM first uses YOLOv5 object detection to draw detection boxes for different objects. Figure 7a is the incoming ordinary RGB image, and Figure 7b is the image after YOLOv5 processing. As can be seen, YOLOv5 drew detection boxes for detecting objects of different categories and determined the types of objects to which they belonged.

Then, AHY-SLAM classifies different types of objects according to dynamic levels and divides all the objects into high-dynamic objects, low-dynamic objects, and static objects. For example, both people and vehicles belong to high-dynamic objects, which are the first part of the image in which to eliminate feature points. Mice and cups belong to low-dynamic objects, and in most cases, the feature points need to be retained for tracking. Computers, desks, and so on are static objects, and the feature points can be directly regarded as static feature points. All detection box information and object category information is output to non-keyframes with FAST feature points and keyframes with ORB feature points. According to this information, AHY-SLAM determines whether corners and feature points belong to static points. If they are dynamic points, they need to be eliminated. The specific methods are as follows: For highly dynamic objects, dynamic points in the images processed using either the optical flow method or the ORB feature point method are eliminated. For low-dynamic objects, only dynamic points in keyframes are eliminated, while non-keyframes retain this part. For static objects, all feature points are directly retained. Because the detection box in YOLO is rectangular, if all feature points in the dynamic object box are directly eliminated, a large number of static points will be mistakenly deleted. Therefore, according to the above classification rules, the overlapped part of the detection boxes is determined. If the dynamic object box overlaps the static object box, all feature points in the static object box are retained first. Then, the remaining feature points in the dynamic object box are eliminated.**Algorithm** **3:** Object detection based on YOLOv5 and dynamic feature point elimination process
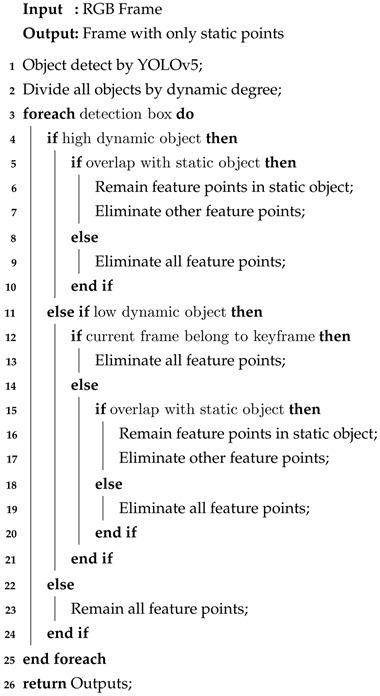


The result of eliminating the abnormal points on the dynamic object is shown in Figure 8a, which is the image after extracting the feature points, and Figure 8b is the image after removing the dynamic feature points. It can be seen that in Figure 8b, the feature points in the overlapping part between the high-dynamic object “man” and the static object “computer” are retained according to the static feature points, while all the red dynamic points on the dynamic object are eliminated.

## 4. Experiments and Analysis

### 4.1. Experimental Setup

In order to test the performance of AHY-SLAM, the TUM dataset was used for the experimental evaluation, and a quantitative comparison test was conducted with the current mainstream SLAM methods. The TUM dataset was produced by the University of Munich in Germany, using the Kinect RGB-D camera to capture information at a rate of 30 Hz, with an image resolution of 640 × 480. The dataset provides real-time camera position and attitude information, which can be approximated to the real position data on the RGB-D camera. In this paper, eight dynamic scene sequences from the TUM RGB-D dataset were mainly used for the experiments, which were divided into walking and sitting positions. The comparison objects were ORB-SLAM2, DS-SLAM, and DO-SLAM.

The experimental environment was configured with Ubuntu 20.04, GeForce RTX 3070 graphics card, 8 GB video memory, 6-core AMD Ryzen5 3600 CPU, and 16 GB RAM.

### 4.2. Experimental Results

Since AHY-SLAM is an improvement based on ORB-SLAM, the experimental results of AHY-SLAM and ORB-SLAM2 are first compared. In this paper, the absolute trajectory error (ATE) and the relative pose error (RPE) were used to evaluate the accuracy of SLAM.ATE, wherein the ATE represents the direct difference between the pose estimate and the ground truth, and the RPE represents the relative translation error and relative rotation error derived from odometry measurements. Three data including the root-mean-square error (RMSE), mean error (mean), and standard deviation (Std) of the ATE and RPE were selected for comparison [45].

As can be seen in Table 2 and Table 3, compared with the original ORB-SLAM, AHY-SLAM is significantly improved in dynamic scenes. For sequences with more dynamic objects in the image and a greater motion degree, the accuracy of AHY-SLAM is higher, and the ATE can be reduced by up to 97%.

Figure 9 and Figure 10 show the distribution graphs of the absolute trajectory errors of AHY-SLAM and ORB-SLAM2 for the walking_xyz and walking_half sequences, respectively, where Figure 9a and Figure 10a is the ATE of AHY-SLAM, and Figure 9b and Figure 10b is the ATE of ORB-SLAM2.

Figure 11 and Figure 12 show the absolute trajectory error graphs of AHY-SLAM and ORB-SLAM2 in three-dimensional space for the walking_xyz and walking_half sequences, respectively, where Figure 11a and Figure 12a is the ATE of AHY-SLAM, and Figure 11b and Figure 12b is the ATE of ORB-SLAM2.

In order to make the comparison effect more obvious, we put the estimated trajectory of AHY-SLAM, the estimated trajectory of ORB-SLAM2, and the real trajectory together for comparison. Figure 13 shows the comparison graph of the three trajectories in three-dimensional space, and Figure 14 shows the comparison graph of the three trajectories in the directions of xyz and rpy, where Figure 13a and Figure 14a is the walking_xyz sequence, and Figure 13b and Figure 14b is the walking_half sequence. In the figure, the black lines represent the real trajectories, the blue lines represent the AHY-SLAM estimated trajectories, and the orange lines represent the ORB-SLAM2 estimated trajectories.

In all the results shown above, it is clear that AHY-SLAM has a significant advantage over ORB-SLAM2 in terms of the accuracy of pose estimation in dynamic scenes. The estimated trajectory of ORB-SLAM2 is far from the real trajectory when facing the dynamic scene, but AHY-SLAM can complete the trajectory estimation in the dynamic scene well.

In order to further verify the superiority of AHY-SLAM, DS-SLAM and Detect-SLAM were selected in this paper for comparative experiments. In these experiments, the RMSE, mean, and std values of the absolute trajectory errors were selected as comparison indexes for verification. The experimental results are shown in Table 4. The effect of AHY-SLAM is better than that of DS-SLAM and Detect-SLAM in sequences with a large motion amplitude of dynamic objects. However, when the dynamic objects determined with AHY-SLAM move less, the effect of DS-SLAM is better. This is because AHY-SLAM does not take into account that feature points on dynamic objects may remain static, but instead directly eliminates all possible moving feature points, resulting in a small number of remaining static points, which affects the accuracy of tracking and positioning.

Real-time performance is also one of the important evaluation indexes for SLAM. SLAM that blindly pursues accuracy and loses real-time performance is difficult to apply to realistic scenes. In order to verify the real-time performance of AHY-SLAM, this paper selected DS-SLAM, Detect-SLAM, DynaSLAM, RDS-SLAM [46], and ORB-SLAM2 to compare the time used to process each frame in different sequences. The experimental results are shown in Table 5.

As can be seen in the above table, AHY-SLAM is more real-time than other dynamic scene SLAM methods, and it takes more time than ORB-SLAM2 because of the added object detection process, but it still operates at a lower level. Our approach can also be combined with related approaches to hardware acceleration, which can be used to further improve processing performance in performance-sensitive scenarios [47]. Visual attention can also be introduced into YOLOv5 to accelerate the recognition of dynamic objects [48,49].

## 5. Discussion

In this paper, for dynamic scenes, based on ORB-SLAM2, we proposed a new SLAM method, AHY-SLAM, which combines the LK optical flow method and adaptive threshold homogenization feature extraction and uses YOLOv5 to eliminate dynamic feature points. By comparing it with other dynamic scene SLAM methods in the dynamic sequences of the TUM data set, we can conclude the following:

(1) Compared with the original method of ORB-SLAM2, the performance of AHY-SLAM is significantly improved in dynamic scenes. The more dynamic the targets in the scene and the larger the target motion range, the more obvious the advantage of AHY-SLAM is, and the positioning accuracy is improved by more than 50% in multiple sequences.

(2) Compared with DS-SLAM and Detect-SLAM, which are also designed for dynamic scenes, the positioning accuracy of AHY-SLAM in all dynamic sequences is higher than that of Detect-SLAM. AHY-SLAM also performs better than DS-SLAM in sequences with a large motion amplitude of dynamic targets.

(3) In comparison with the real-time performance of the system, AHY-SLAM is significantly faster than DS-SLAM, RDS-SLAM, and DynaSLAM. AHY-SLAM also has a speed advantage over Detect-SLAM, which also uses object detection.

By summarizing the above performance of AHY-SLAM, it can be concluded that compared with ORB-SLAM, this method is greatly improved in its applicability in dynamic scenes, and compared with other dynamic scene SLAM methods, it not only improves positioning accuracy but also satisfies real-time requirements.

## 6. Conclusions

In this paper, we proposed a new dynamic scene SLAM method: AHY-SLAM. Compared with ORB-SLAM2, AHY-SLAM has three new modules: a keyframe selection module, a threshold calculation module, and an object detection module. In addition, homogenized ORB feature point extraction and dynamic feature point elimination steps are added to the tracking thread. AHY-SLAM uses the optical flow method to reduce the calculation amount for descriptors in the selection of keyframes and extract more uniform feature points that are more suitable for dynamic scenes, and it has the ability to deal with high-dynamic scenes. Compared with ORB-SLAM2, AHY-SLAM has better pose estimation accuracy, which can reduce the absolute trajectory error by up to 97% in the TUM dataset. Compared with other dynamic scene SLAM methods, the speed of AHY-SLAM is also significantly improved under a guarantee of acceptable accuracy. In future work, we will continue to improve the judgment process of AHY-SLAM for dynamic points, try to add threads to construct dense semantic maps according to existing semantic map construction methods, and use semantic maps to realize path planning in dynamic scenes.

## Figures and Tables

**Figure 1 sensors-23-04241-f001:**
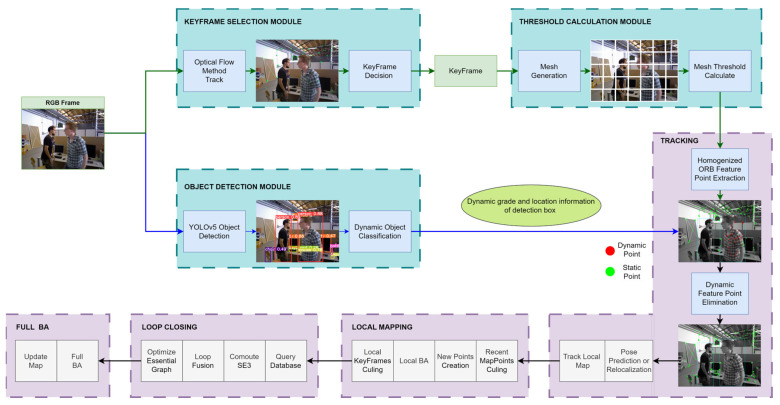
The system architecture of AHY-SLAM. The blue box is the innovative part of AHY-SLAM compared with ORB-SLAM2.

**Figure 2 sensors-23-04241-f002:**
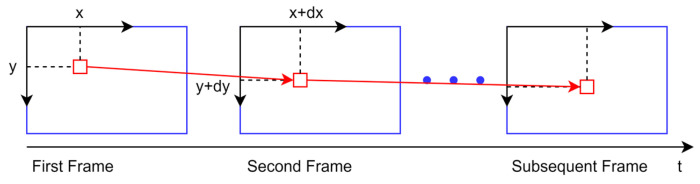
LK optical flow tracking diagram. The red boxes in the figure represent the position of a FAST corner in different frames.

**Figure 3 sensors-23-04241-f003:**
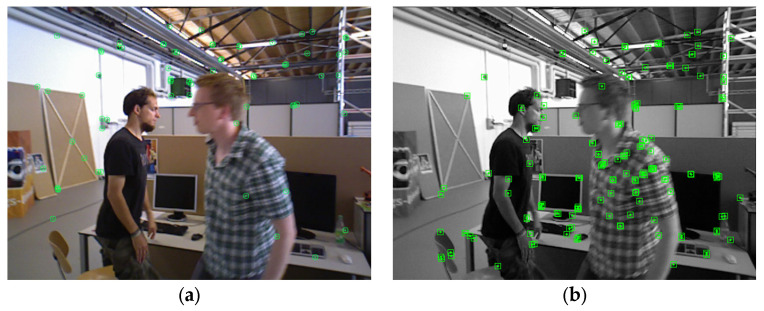
When the optical flow method fails to track enough corners, the ORB feature point method is used to re-extract corners. Green marks are traced corners. (**a**) Optical flow method to track the number of corners is insufficient; (**b**) ORB feature point method re-extracts corners.

**Figure 4 sensors-23-04241-f004:**
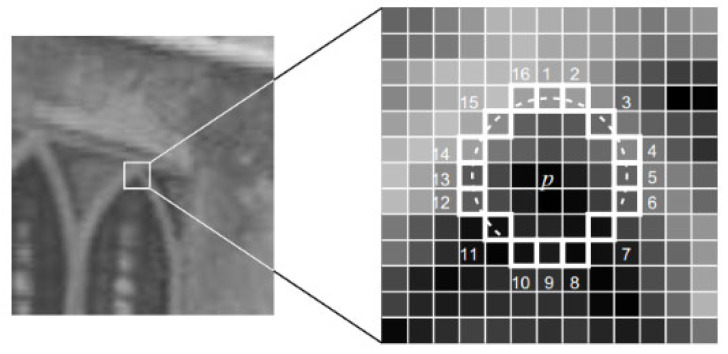
Bresenham circle in FAST corner extraction. The 16 pixels on the Bresenham circle are numbered.

**Figure 5 sensors-23-04241-f005:**
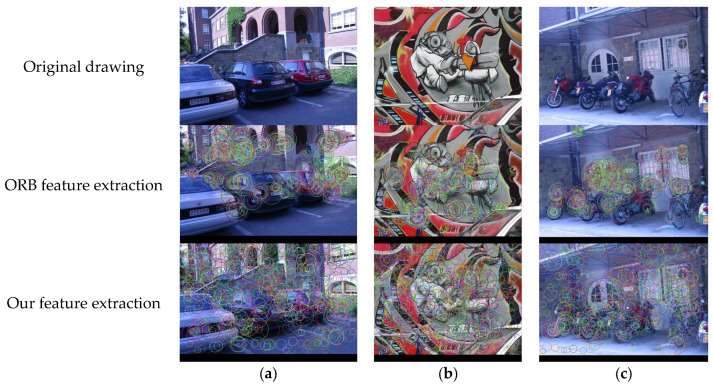
Our feature extraction method and the traditional ORB algorithm compared in the Mikolajczyk open-source image dataset. (**a**) Cars image; (**b**) doodle image; (**c**) bikes image.

**Figure 6 sensors-23-04241-f006:**
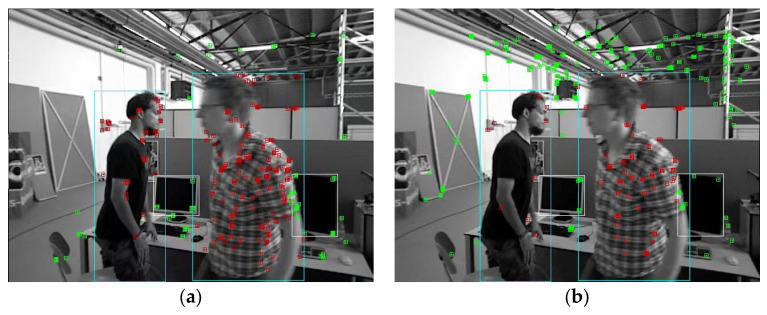
ORB feature points extracted from dynamic scenes. (**a**) is the processing result without using homogenized extraction; (**b**) is the processing result after using the homogenized extraction method in this paper. The red points are dynamic feature points, and the green points are static feature points. The light green boxes represent dynamic targets, and the white boxes represent static targets.

**Figure 7 sensors-23-04241-f007:**
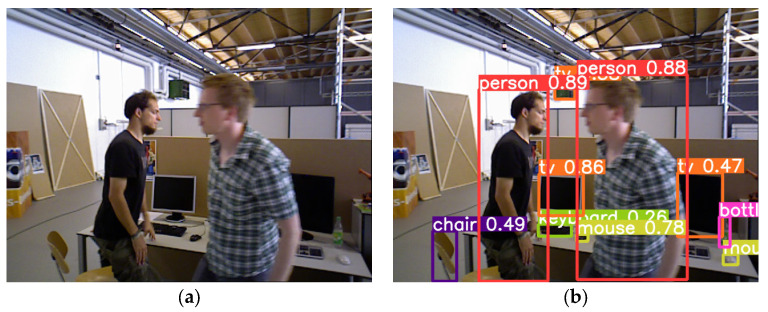
Processed RGB images using YOLOv5 object detection; (**a**) is the incoming ordinary RGB image, and (**b**) is the image after YOLOv5 processing.

**Figure 8 sensors-23-04241-f008:**
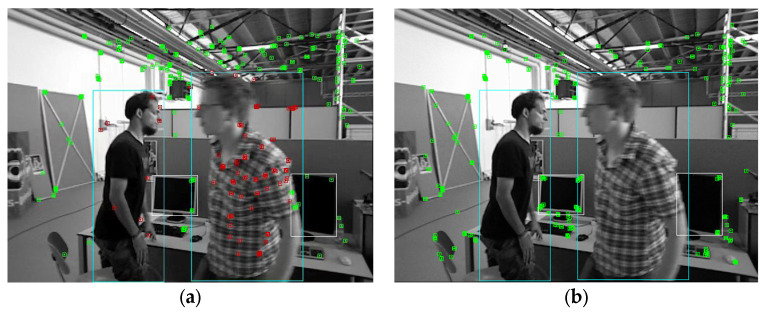
Dynamic feature points were eliminated using yolov5 object detection. (**a**) is the image with all feature points; (**b**) is the image after removing the dynamic feature points. The red points are dynamic feature points, and the green points are static feature points. The light green boxes represent dynamic targets, and the white boxes represent static targets.

**Figure 9 sensors-23-04241-f009:**
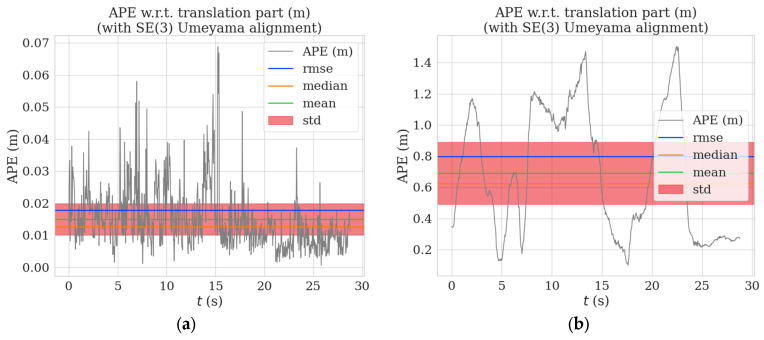
Absolute trajectory error distribution graphs for walking_xyz sequence: (**a**) is the ATE of AHY-SLAM; (**b**) is the ATE of ORB-SLAM2.

**Figure 10 sensors-23-04241-f010:**
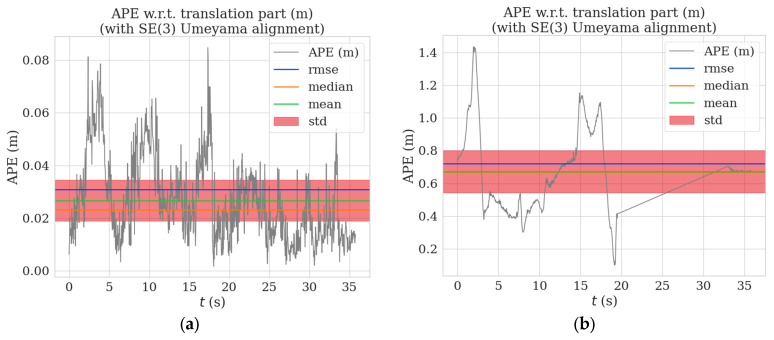
Absolute trajectory error distribution graphs for walking_half sequence: (**a**) is the ATE of AHY-SLAM; (**b**) is the ATE of ORB-SLAM2.

**Figure 11 sensors-23-04241-f011:**
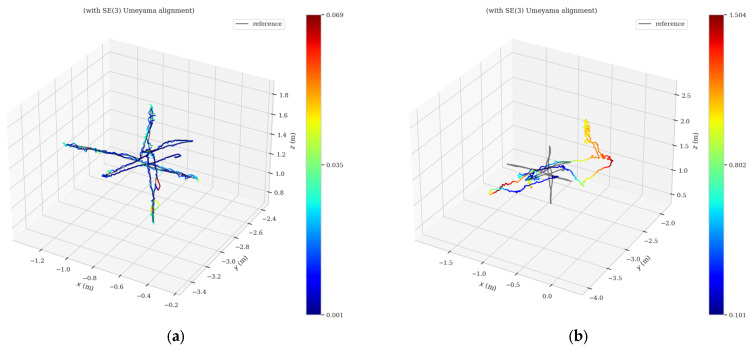
Absolute trajectory error graphs in 3D space for walking_xyz sequence: (**a**) is the ATE of AHY-SLAM; (**b**) is the ATE of ORB-SLAM2.

**Figure 12 sensors-23-04241-f012:**
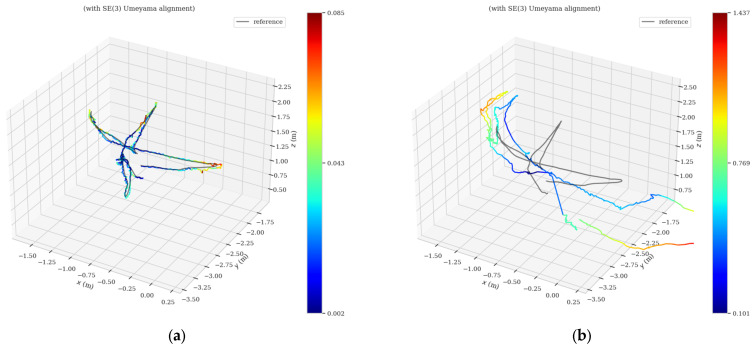
Absolute trajectory error graphs in three-dimensional space for walking_half sequence: (**a**) is the ATE of AHY-SLAM; (**b**) is the ATE of ORB-SLAM2.

**Figure 13 sensors-23-04241-f013:**
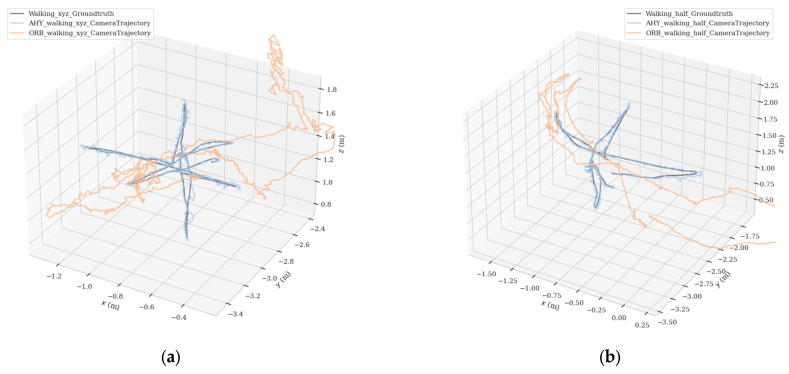
Comparison graph of the three trajectories in three-dimensional space, where (**a**) is the walking_xyz sequence, and (**b**) is the walking_half sequence. In the figure, the black lines represent the real trajectories, the blue lines represent the AHY-SLAM estimated trajectories, and the orange lines represent the ORB-SLAM2 estimated trajectories.

**Figure 14 sensors-23-04241-f014:**
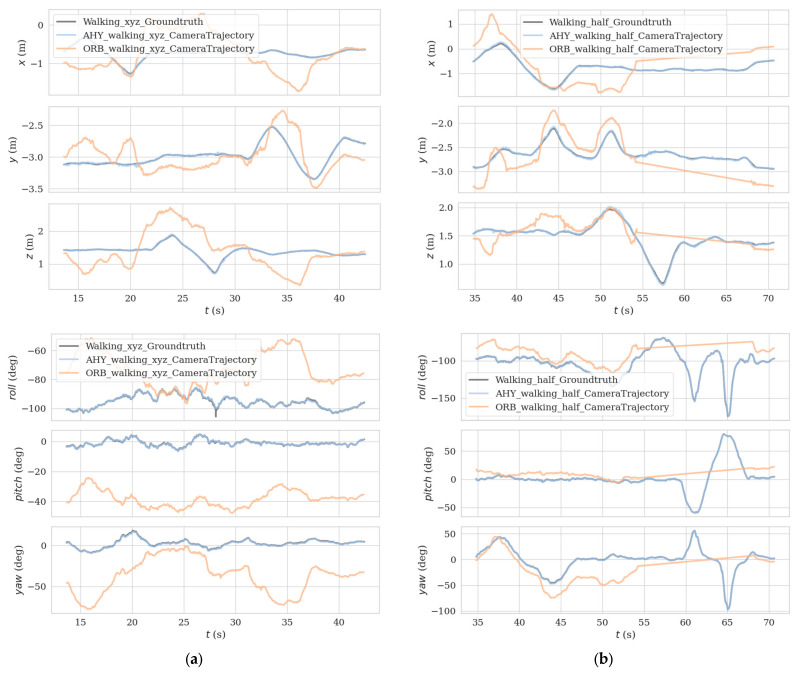
Comparison graph of the three trajectories in the directions of xyz and rpy, where (**a**) is the walking_xyz sequence, and (**b**) is the walking_half sequence. In the figure, the black lines represent the real trajectories, the blue lines represent the AHY-SLAM estimated trajectories, and the orange lines represent the ORB-SLAM2 estimated trajectories.

**Table 1 sensors-23-04241-t001:** Comparison of uniformity between our proposed method and ORB method.

Image	Cars	Doodle	Bikes	Boat	Bark
ORB method	192.33	202.36	201.54	196.25	198.65
Our method	145.56	142.68	165.24	146.23	155.45
Improvement	24.3%	29.5%	18.0%	25.5%	21.7%

**Table 2 sensors-23-04241-t002:** Comparison of ATE between ORB-SLAM2 and AHY-SLAM in TUM sequences.

Sequence	ORB-SLAM2/m			AHY-SLAM/m			Improvement/%		
	RMSE	Mean	Std	RMSE	Mean	Std	RMSE	Mean	Std
walking_xyz	0.7978	0.6903	0.3999	0.0178	0.0150	0.0096	97.77	97.83	97.60
walking_half	0.7202	0.6721	0.2587	0.0309	0.0267	0.0156	95.71	96.03	93.97
walking_static	0.3687	0.3328	0.2524	0.0074	0.0072	0.0033	97.99	97.84	98.69
waking_rpy	1.6429	1.5375	0.6291	0.1843	0.1495	0.1538	88.78	90.28	75.55
sitting_xyz	0.0149	0.0129	0.0075	0.0136	0.0115	0.0067	8.73	10.85	10.67
sitting_half	0.0486	0.0424	0.0377	0.0178	0.0166	0.0152	66.37	60.85	59.68
sitting_static	0.0158	0.0147	0.0068	0.0081	0.0073	0.0047	48.73	50.34	30.88
sitting_rpy	0.0491	0.0355	0.0301	0.0425	0.0323	0.0282	13.44	9.01	6.31

**Table 3 sensors-23-04241-t003:** Comparison of RPE between ORB-SLAM2 and AHY-SLAM in TUM sequences.

Sequences	ORB-SLAM2/m			AHY-SLAM/m			Improvement/%		
	RMSE	Mean	Std	RMSE	Mean	Std	RMSE	Mean	Std
walking_xyz	0.0291	0.0234	0.0173	0.0133	0.0105	0.0081	54.30	55.13	53.18
walking_half	0.0308	0.0222	0.0215	0.0141	0.0116	0.0079	55.22	47.75	63.26
walking_static	0.0325	0.0211	0.0258	0.0079	0.0071	0.0065	75.69	66.35	74.81
waking_rpy	0.0282	0.0229	0.0182	0.0258	0.0211	0.0161	8.51	7.86	11.54
sitting_xyz	0.0121	0.0099	0.0052	0.0086	0.0077	0.0043	28.93	22.22	17.31
sitting_half	0.0196	0.0113	0.0181	0.0152	0.0102	0.0117	22.45	9.74	35.36
sitting_static	0.0156	0.0130	0.0052	0.0082	0.0077	0.0043	47.44	40.77	17.31
sitting_rpy	0.0212	0.0181	0.0129	0.0201	0.0146	0.0117	5.19	19.34	9.30

**Table 4 sensors-23-04241-t004:** Comparison of absolute trajectory errors between AHY-SLAM and other dynamic scene SLAM methods. The bold represents optimal results.

Sequence	DS-SLAM/m		Detect-SLAM/m		AHY-SLAM/m	
	RMSE	Std	RMSE	Std	RMSE	Std
walking_xyz	0.0266	0.0175	0.0246	0.0125	**0.0178**	**0.0096**
walking_half	0.0319	0.0161	0.0521	0.0144	**0.0309**	**0.0156**
walking_static	0.0092	0.0046	0.0162	0.0034	**0.0074**	**0.0033**
waking_rpy	0.4217	0.2612	0.2951	0.2325	**0.1843**	**0.1538**
sitting_xyz	0.0179	0.0122	0.0211	0.0079	**0.0136**	**0.0067**
sitting_half	**0.0154**	**0.0068**	0.0234	0.0170	0.0178	0.0152
sitting_static	**0.0071**	**0.0029**	0.0085	0.0041	0.0081	0.0047
sitting_rpy	**0.0258**	**0.0149**	0.0486	0.0376	0.0425	0.0282

**Table 5 sensors-23-04241-t005:** Comparison of real-time performance between AHY-SLAM and other dynamic scene SLAM methods.

Dataset	DS-SLAM	Detect-SLAM	DynaSLAM	RDS-SLAM	ORB-SLAM2	AHY-SLAM
walking_xyz	0.757	0.332	1.116	0.203	0.0375	0.0896
walking_half	0.683	0.345	1.154	0.212	0.0328	0.0968
walking_static	0.712	0.328	1.131	0.209	0.0391	0.0913

## Data Availability

The research data used in this paper are from https://vision.in.tum.de/data/datasets/rgbd-dataset/download (accessed on 1 December 2022).

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
