# Peer review of "AHY-SLAM: Toward Faster and More Accurate Visual SLAM in Dynamic Scenes Using Homogenized Feature Extraction and Object Detection Method"

_sensors, 2023, doi:10.3390/s23094241_

Round 1

Reviewer 1 Report

This paper presents a method for visual SLAM.

The studied topic is interesting and also meaningful.

The paper still has some major problems.

The authors are suggested to revise the paper given the following comments.

More closely-related methods are suggested to be compared.

More large-scale open datasets are also suggested to be used for performance comparison.

As discussed in some surveys and studies (‘Perceptual image quality assessment: a survey’, ‘Screen content quality assessment: overview, benchmark, and beyond’, ‘Study of subjective and objective quality assessment of audio-visual signals’, image/video quality is an important aspect of various intelligent systems. High-quality images/videos are important for the successful usage of these intelligent systems, while low-quality media may degrade the performance of these systems.

The authors may give some discussions on this aspect and the above mentioned works.

Feature point is an important aspect of the proposed work. It is also widely used in various computer vision applications, for example image quality assessment (‘Blind quality assessment based on pseudo-reference image’, ‘Blind image quality estimation via distortion aggravation’, ‘Unified blind quality assessment of compressed natural, graphic, and screen content images’). The authors are suggested to give some literatures in the paper, for better referring of the relevant topics.

The writing of the paper still needs improvement.

Reviewer 2 Report

This paper discusses proposed a dynamic scene visual SLAM, AHY-SLAM, based on homogenized feature extraction and object detection method. keyframe selection module, threshold calculation module, and object detection module were added in AHY-SLAM. The original operating framework of ORB-SLAM2 is modified based on a new keyframe selection method on the basis of LK optical flow. The method is tested using TUM dataset. AHY-SLAM is also significantly improved under the guarantee of acceptable accuracy. Thus, I suggest the paper can be accepted in current form.

Author Response

Thank you very much for your recognition of our articles, your support is very important to us, wish you a happy life.

Reviewer 3 Report

It is an interesting article with a clear description of the research need; however, it requires a few improvements before publication. The paper describes new solutions in SLAM based on the threshold adaptive homogenized feature extraction and YOLOv5 object detection network with verification on the publicly available dataset. The experiment shows promising effects of the proposed method, but it is still a few considerations (minor revision) that need to be addressed before the publication of the article:

1.       It is necessary to add a list of abbreviations due to the presence of a large number of them in the article.

2.       It is advisable to add the abbreviation AHY - in the abstract and in the introduction to the article.

3.       Fig. 1 - it is necessary to improve the readability of graphics and texts.

4.       The article mentions the Mikolajczyk open-source image dataset twice - the relevant reference and reference in the bibliography are not included.

5.       The caption of Fig. 5 - What is “Leuven image”?

6.       Fig. 9, Fig. 3 - the size of the fonts in the drawings should be increased and standardized to improve readability.

7.       In Fig. 13 the black line is too weakly visible.

8.       In Fig. 14 the black line is too weakly visible.

9.       In Fig. 14 there is an incomprehensible entry "+1.3418...........".

10.   According to the reviewer, it is advisable to add a "Discussion" section (or subsection) to the article, which can be partly separated from the "4.2. Experiment Results" subsection. The purpose of the discussion is to interpret and describe the significance of your findings in light of what was already known about the research problem being investigated, and to explain any new understanding or fresh insights about the problem after you've taken the findings into consideration.  The discussion should clearly explain how your study has moved the reader's understanding of the research problem forward from where you left them at the end of the introduction.

Reviewer 4 Report

AHY-SLAM: Towards Faster and More Accurate Visual SLAM in Dynamic Scenes by Homogenized Feature Extraction and Object Detection Method

In this manuscript, the authors propose a new dynamic scene visual SLAM based on threshold adaptive homogenized feature extraction and YOLOv5 object detection, named AHY-SLAM. AHY-SLAM adds three new modules based on ORB-SLAM2: keyframe selection module-threshold calculation module and object detection module. Moreover, the homogenized ORB feature point extraction and dynamic feature point elimination steps are added to the tracking thread. Every frame entered into AHY-SLAM will be filtered by the optical flow method in the keyframe selection module, and the keyframes will be entered into the threshold calculation module to calculate the threshold values of different regions, these thresholds are used for more homogenized ORB feature point extraction. Compared with ORB-SLAM2, AHY-SLAM has significantly improved pose estimation accuracy over multiple dynamic scene sequences in TUM open dataset. the absolute pose estimation accuracy can be increased by up to 97%. Compared to other dynamic scene SLAM, The speed of AHY-SLAM is also significantly improved under the guarantee of acceptable accuracy.

This research is interesting and it is well presented. However, I have observed some issues that should be addressed in order to accept this manuscript. These are the following ones:

§  The abstract is particularly large and needs to summarize better the research results.

§  I do not particularly like how the abstract and even the introduction start: “SLAM cannot distinguish dynamic feature points from static feature points in the face of dynamic scenes. In the matching process, dynamic points will be incorrectly added to the pose calculation of the camera, resulting in low precision and poor robustness of pose estimation.” I think the authors should make an extra effort to contextualize the problem in a broader context that allows everyone to understand the importance of their research.

§  The excessive use of reiterative expressions that must be reduced for improving the readability and clarity. For example, “(…) YOLOv5 object detection, named AHY-SLAM. AHY-SLAM adds three new modules (…)” or some lineas ahead “To solve the above problem (1), based on the original ORB feature detection algorithm, this paper proposes a threshold adaptive homogenized feature extraction algorithm”.

§  Some expressions must be improved. For example, “But most SLAM is based on static scenes, not optimized for dynamic scenes.” -> The authors should be more clear arguing something like: “Most of the SLAM-algorithms were developed for static scenes and therefore showing worst performance on dynamic scenes/environments”.

§  The authors argue “If the dynamic object has strong texture information, the system will extract a large number of features from the dynamic object.” -> They should show how this problem emerged in the past and how was solved (literature review). Some lines ahead the authors say “Compared to other dynamic scene SLAM, …” -> Which ones?

§  The authors refer to problems and contributions by using bracket and number form-> Just for example, “To solve the above problem (1), based on …. and after that “The main contributions of this paper are as follows: (1) A dynamic scene visual SLAM…” -> The authors should be more consistent with the use of brackets and to think in other way to use bullets/numerations.I have observed how some words are in initial capital letter and other that should be they are not. -> For example, “Compared to other dynamic scene SLAM, The speed of AHY-SLAM is also significantly improved under the guarantee of acceptable accuracy”. In the abstract happens the opposite “scene sequences in TUM open dataset. the absolute pose estimation accuracy can be increased by up to 97%.” -> The authors should revise the whole paper and avoid this kind of issues.

Round 2

Reviewer 1 Report

Most of the previous concerns have been addressed. Some further minor comments are as follows. As described in the literatures, for example, Fixation prediction through multimodal analysis, attention can be of great value in various computer vision applications. The authors may give some discussions on whether incorporating visual attention can further improve the proposed method. 

The studied topic visual SLAM through object detection is very close to the barcode detection in images, as discussed in the following papers: A Multimodal Saliency Model for Videos With High Audio-Visual Correspondence. The above paper also discuss small object detection in wild images. The authors may also give some discussions on these works.
